ecology/physiology/environmental science

artificial light, diapause, flesh fly, photoperiodism, urbanization, urban warming

**Author for correspondence:**
Ayumu Mukai
e-mail: ayumu.mukai@lif.setsunan.ac.jp

# Urban warming and artificial light alter dormancy in the flesh fly

Ayumu Mukai[1,2], Koki Yamaguchi[1] and Shin G. Goto[1]

[1]Department of Biology and Geosciences, Graduate School of Science, Osaka City University, Osaka, Japan
[2]Department of Life Science, Faculty of Science and Engineering, Setsunan University, Osaka, Japan

AM, 0000-0003-1836-071X; SGG, 0000-0002-4431-7531

Seasonal changes in temperature and day length are distinct between rural and urban areas due to urban warming and the presence of artificial light at night. Many studies have focused on the impacts of these ubiquitous signatures on daily biological events, but empirical studies on their impacts on insect seasonality are limited. In the present study, we used the flesh fly *Sarcophaga similis* as a model insect to determine the impacts of urbanization on the incidence and timing of diapause (dormancy), not only in the laboratory but also in rural and urban conditions. In the laboratory, diapause entry was affected by night-time light levels as low as 0.01 lux. We placed fly cages on outdoor shelves in urban and rural areas to determine the timing of diapause entry; it was retarded by approximately four weeks in urban areas relative to that in rural areas. Moreover, almost all flies in the site facing an urban residential area failed to enter diapause, even by late autumn. Although an autumnal low temperature in the urban area would mitigate the negative effect of artificial light at night, strong light pollution seriously disrupts the flesh fly seasonal adaptation.

## 1. Introduction

The conversion of natural and semi-natural habitats to urban areas increased drastically worldwide due to rapid demographic and economic growth [1,2]. The ecological effects of urbanization are initially realized through substantial changes in the physical environment, including the atmosphere, water bodies, temperature and land surface, resulting in a marked urban–rural difference [3,4]. These changes affect urban ecosystems and their components at the individual, population and community levels [5,6]. The most common approach to investigating the ecological consequences of urbanization involves the analysis of species abundance, species diversity and species interactions [7–10]. The effect of urbanization on individuals is, however, necessarily mediated through behavioural and physiological responses [11,12].

Two of the most ubiquitous signatures of urban environments are urban warming and artificial light at night [3,11,13]. Urban warming can produce increases of 5–9°C in surface temperature [14]. Species with higher critical thermal optima tend to be better adapted to survive in urban areas than those with lower critical thermal optima. Consequently, species with lower optima experience population declines in urban areas [15]. Urban warming also alters species phenology in urban areas relative to that in nearby undeveloped rural areas [16,17]. Artificial light at night alters natural light cycles, the strongest and most predictable environmental signals that organisms typically rely on [13]. Artificial night-time light impacts multiple daily events including diel behaviour/activity, predator–prey interactions, temporal niche partitioning and sleep [18–23]. It also impacts seasonal events including growth, reproductive behaviour and output, migration, flowering, vegetation and biomass [19,24–32].

Insects are small and their size allows them to use a wide range of niches in most environments. However, their small size also makes them vulnerable. Because they are sensitive to seasonal temperature changes and starvation, they must anticipate the arrival of the seasons by sensing environmental cues to prepare physiologically and/or behaviourally. Temperate insects anticipate seasonal changes by monitoring day length and exhibit a biological response, known as photoperiodism, to these variations [33,34]. For example, the flesh fly, Sarcophaga similis (Meade) develops without interruption during the long days of summer. By contrast, it enters diapause at the pupal stage in response to shorter autumn days [35–38]. Diapause is a hormonally regulated developmental arrest and is accompanied by a major metabolic shutdown [39]. Insects destined to enter diapause accumulate additional energy reserves; these reserves and a limited metabolic rate during diapause allow insects to survive harsh environmental conditions [40]. Temperature is also a useful cue for timing periodic behaviours and triggering physiological changes that affect reproductive success. However, the temperature can be unreliable due to large daily fluctuations and inter-annual variation, particularly in terrestrial environments; thus, insects use temperature as a supplemental cue in photoperiodism [41].

Urbanization is considered to affect the timing of diapause entry theoretically [13,42]. However, little attention has been paid to this phenomenon and very few studies have focused on the effect of urban warming on the timing of diapause entry. Moreover, only a few studies have focused on the effect of artificial light at night. Westby & Medley [43] found that simulated urban artificial light at night (illumination at 9–11 lux) reduces diapause incidence by as much as 40% in a species of mosquito [43]. However, in this experiment, adult mosquitoes had been moved to the laboratory to stimulate mating, feeding and egg-laying at laboratory rearing temperatures; thus, the negative effect of light illumination may have been underestimated. The study of van Geffen et al. [44] illuminated autumn nights with red, green or warm-white light-emitting diodes (LEDs) to obtain intensities of 7.0 lux in the field [44]. Control cabbage moth pupae under no illumination at night effectively entered diapause, whereas most under white and green light, and approximately 20% of pupae under red light, averted diapause. Fyie et al. [45] also found in a mosquito that dim light (approx. 4 lux) at the dark phase of short-day conditions averted diapause entry [45]. However, these studies only assessed the deleterious effects of light at night under 'simulated' light conditions at a fixed illuminance. Studies focusing on the effect of urbanization on diapause entry under field conditions are scarce.

Insects are the most diverse group of animals and are ubiquitous across terrestrial ecosystems, playing key ecological roles [46,47]. However, they are rapidly declining worldwide [48] and urbanization is one of the major drivers [10]. We still have limited information on its impact on insects, especially on their seasonal adaptation strategy. In the present study, we focused on the effect of urban warming and artificial light at night on diapause entry in a flesh fly. Recognizing how these factors affect insect seasonality would greatly help to mitigate their negative effects.

# 2. Material and methods

## 2.1. Insects

Colonies of S. similis were obtained from females captured in Osaka City (34.59° N, 135.50° E) and Toyonaka City (34.80° N, 135.45° E), Japan, in 2014 and 2018, respectively. Stock cultures were maintained under diapause-averting long-day conditions (16 h light/8 h dark) at 25 ± 1°C. Fluorescent light (FL15SW, FL20SW, FL30SW or FL40SW; Panasonic, Kadoma, Japan) was used as the light source during the light phase (greater than 1000 lux). Water, two sugar blocks and a piece of beef or chicken liver were provisioned for adult flies in a plastic container (15 cm in diameter, 9 cm in depth) covered

with nylon netting. A second, fresh piece of liver was provided as a larviposition site 12 days after the first provision. One day later, the livers with larvae were collected and transferred to an aluminium dish. The dish was placed on dry wood chips (30–50 mm in depth) as a pupariation/pupation substrate in a plastic container. Mature larvae ceased feeding and moved into the wood chips. A few days later, they pupariated and pupated. New adults emerged approximately 18 days after larviposition.

For experiments, newly emerged adults were kept under short-day conditions (12 h light/12 h dark) at 20°C since the first provision of the liver. Twelve days after the provision of the liver, 5–10 flies were transferred to each experimental condition (see below). Three days later, that is 15 days after the first provision of the liver, pharate larvae in the uterus were collected by dissecting the female abdomen, to set the larviposition timing. The photoperiod-sensitive stage of this species range from the embryonic stage in the uterus of the mother to the end of the larval stage and no maternal effect via the uterus induces diapause in offspring [35]. Thus, this larval collection does not affect diapause induction. Larvae collected were transferred to a piece of liver and maintained under the same conditions. Diapause status and sex were assessed 20 and 30 days after larval collection at 20°C and 15°C, respectively, in the laboratory diapause induction experiments, as described in the previous studies [49,50]. Diapause status and sex were assessed 20–30 days after larval collection in semi-natural conditions. The fly pupa is encased in a hardened larval cuticle (puparium). We opened the puparium and judged pupal mortality in the semi-natural experiments.

## 2.2. Laboratory diapause induction

Sarcophaga similis diapause incidence was measured under long-day, short-day and light-at-night short-day (LAN; 12 h light/12 h dim light) conditions at 20 and 15°C. In all conditions, flies were illuminated by fluorescent light (greater than 1000 lux) during the light phase as described above. For the long-day and short-day conditions, no light was provided during the dark phase (0 lux). For the LAN condition, a fluorescent light (FL8WF, Panasonic, Osaka, Japan), covered with a black vinyl sheet to adjust the illuminance level to 0.01, 0.1 or 1 lux, was set in the incubator and left constantly on; thus, in the LAN conditions, the light phase was illuminated by both high and low illuminance, while the dark phase had a single fluorescent light with low illuminance. The spectrum of the fluorescent light measured by the colour rendering illuminometer CL-70F (Konica Minolta, Tokyo, Japan) is shown in electronic supplementary material, figure S1. The LAN conditions with 0.01, 0.1 and 1 lux dim light are abbreviated as LAN-0.01, -0.1 and -1, respectively. We chose these parameters as these illuminance levels are typical in urban areas [42] and the temperatures represented average temperatures of early October, when the flies begin to enter diapause, and early November, when most flies have entered diapause, in our field location in Osaka, Japan (Japan Meteorological Agency, www.jma.go.jp).

## 2.3. Semi-natural diapause induction

In the first experiment (Experiment 1), we compared diapause timing between rural and urban sites. In the second (Experiment 2), we compared diapause timing between two urban sites, one of which was less exposed to artificial light sources and the other was subject to severe light pollution. Illuminance and temperature at each site were recorded by data loggers (TR-74 Ui, T & D Corporation, Nagano, Japan) every 2 min during the experimental period.

In Experiment 1, we placed outdoor shelves (90.7 cm width, 45.7 cm depth, 90.0 cm height), covered by a clear vinyl sheet to shelter from rainfall, at rural and urban sites in 2015. We selected the Botanical Gardens of Osaka City University (BG; 34.77° N, 135.68° E) as the rural site, as no or few artificial light sources existed. We selected the campus of Osaka City University (OCU; 34.59° N, 135.50° E) as the urban site. This urban site faced the urban residential area but was covered by a tree branch; thus, it was partially protected from artificial light sources. Shelves at both sites faced north and were protected from direct sunlight. Adult females that had been reared for 12 days under short-day conditions with the liver in the laboratory were placed on the outdoor shelves every week. Three days after the adult placement, larvae were collected and were continuously kept on the outdoor shelves. The data logger was placed on the shelf. In this experiment, we did not obtain data on mortality and diapause incidence on 2 and 9 November 2015, as we were unable to check survival and diapause status within 30 days of the larval collection. No assessment of sex was conducted in this experiment.

In Experiment 2, we established two sites at OCU, approximately 130 m apart. At site 1, the outdoor shelf was established in the same manner as in 2015 but was placed in a new location due to operational unavailability. This new location was more exposed to residential artificial light sources than was the

OCU urban site in Experiment 1. At site 2, we placed insect-rearing containers in a clear plastic container attached to the branch of a cherry blossom tree. This container faced north in a residential area near a 24 h convenience store; thus, it was exposed to light even during the night. The data logger was placed in the container to monitor temperature and illuminance. Adult females were placed at these locations every week and collected larvae were continuously kept at the same locations, as in Experiment 1. The spectrum of the residential light sources in site 2 after sunset is shown in electronic supplementary material, figure S1. The residential light contained the various wavelengths of light and the pattern is similar to the spectrum of the fluorescent light, not that of the LED and the sodium light. The residential light in the site would be mainly from the fluorescent light.

## 2.4. Data analysis

All data were analysed using R v. 3.1.2 [51]. In the laboratory experiment, diapause incidences under laboratory conditions were compared using Tukey-type multiple comparisons for proportions. Sexual differences were compared using Fisher's exact test. In the semi-natural experiments, we calculated the length of the day when illuminance was continuously higher than 0.01, 0.1 and 1 lux from the illuminance data. Astronomical day length was calculated from sunrise and sunset of each day at BG and OCU obtained from the National Astronomical Observatory of Japan website (eco.mtk.nao.ac.jp). Differences between the astronomical day length and the length of the day at which illuminance was continuously higher than each set value were calculated as means ± s.d. The day of larval collection from the mother was regarded as the day for the experimental population. Two-way analysis of variance (ANOVA) was used for the diapause and mortality data after arcsine transformation. No data on environmental parameters were available for 5 and 6 October 2016 due to a typhoon.

# 3. Results

## 3.1. Effects of temperature and light at night on diapause induction in the laboratory

A clear photoperiodic response was observed at both 20°C and 15°C (Tukey-type multiple comparisons for proportions, $p < 0.05$). Diapause was very rare under long-day conditions, whereas it was common under short-day conditions (figure 1). Diapause induction was altered by artificial light at night (figure 1). At 20°C, diapause incidence decreased as artificial light illuminance increased (figure 1a). Diapause in all three LAN conditions was significantly less frequent than that under short-day conditions (Tukey-type multiple comparisons for proportions, $p < 0.05$), indicating that even light at 0.01 lux during night prevents diapause entry. At 15°C, diapause incidences also decreased as artificial light illuminance increased (figure 1b). Diapause incidences under LAN-0.1 and -1 were significantly lower than those under short-day conditions (100%) (Tukey-type multiple comparisons for proportions, $p < 0.05$). In contrast with LAN-0.01 at 20°C, the diapause-preventing effect of a 0.01 lux light at night (LAN-0.01) was undetectable at 15°C (Tukey-type multiple comparisons for proportions, $p > 0.05$). Diapause under short-day conditions and each LAN condition was significantly increased at 15°C over those at 20°C (Fisher's exact test, $p < 0.05$).

We further compared the diapause incidences between males and females in each condition. No sexual bias was detected in the experimental populations ($\chi^2$ test, $p > 0.05$). At 20°C, diapause was more frequent in males under short-day, LAN-1, -0.1 and -0.01 conditions (Fisher's exact test, $p < 0.05$). At 15°C, diapause was more frequent in males in LAN-1 (Fisher's exact test, $p < 0.05$). Diapause was also more frequent in males in LAN-0.1, but the difference was as small as 4%. No sexual difference was detected under conditions in which almost all individuals entered or averted diapause, that is, long-day conditions at 20°C, short-day conditions at 15°C and LAN-0.01 at 15°C (Fisher's exact test, $p > 0.05$).

## 3.2. Temporal changes in environmental parameters under semi-natural conditions

The environmental parameters for the semi-natural conditions are shown in figure 2. In Experiment 1, we compared the urban (OCU) and rural (BG) sites. Daily average temperatures declined over time with some fluctuation at both sites, although the daily temperature at OCU was higher than that of BG by 0.3–6.8°C, due to urban warming (figure 2a). We further calculated 7-day moving average temperatures to observe the pattern of temperature decline (data not shown). These averages revealed

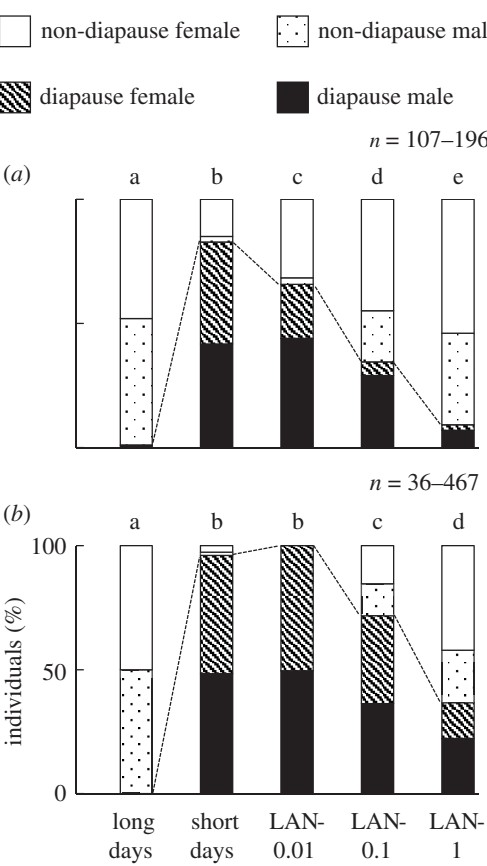

**Figure 1.** Incidences of *S. similis* in diapause and non-diapause under long-day (16 h light/8 h dark), short-day (12 h light/12 h dark) and light-at-night short-day (LAN; 12 h light/12 h dim light) conditions at 20°C (*a*) and 15°C (*b*). Pupae were sexed. LAN conditions with 1, 0.1 and 0.01 lux dark phase are abbreviated as LAN-1, -0.1 and -0.01, respectively. Note that *n* indicates the number of pupae. The different letters in each panel indicate significant differences in diapause incidence (Tukey-type multiple comparisons for proportions, $p < 0.05$).

that the temperature decline at BG predated that at OCU by three to four weeks; the earliest days at which the temperature fell to 17.5 and 15°C at BG were 3 and 25 October 2015, respectively, and those at OCU were 28 October and 22 November 2015, respectively. Temporal changes in the length of the day when illuminance was continuously higher than 1 lux were similar between the two sites (figure 2*b*). These day lengths shortened in parallel to the astronomical day length, and overall, the differences between day lengths at BG and OCU and the astronomical day length were as small as 0.5 or 0.6 h (table 1). When the threshold was set to 0.1 lux, day lengths at BG were 0.25 h shorter than astronomical day lengths; by contrast, those at OCU were 10.7 h longer than astronomical day lengths (figure 2*c* and table 1). The threshold at 0.01 lux also showed a similar result (figure 2*d* and table 1). Therefore, the BG site in the rural area was less polluted by artificial light sources and its nights were darker, whereas the OCU site in the urban area was polluted by artificial light sources and was brighter than 0.1 lux on most nights.

In Experiment 2, daily average temperatures at sites 1 and 2 similarly declined along with the transition of days with fluctuation in a similar manner. The temperature differences between the sites were as small as 0.0–0.9°C (figure 2*g*). It was obvious that site 1 was less, whereas site 2 was severely, exposed to artificial light sources. Illuminance detected at sites 1 and 2 during nights (from sunset to sunrise) were $0.06 ± 0.09$ and $7.07 ± 2.80$ lux, respectively (mean ± s.d.) (electronic supplementary material, figure S2). When the threshold was set at 1 lux, the day length at site 1 was 0.57 h shorter than that of the astronomical day length (figure 2*h* and table 1). By contrast, the day length was 9.20 h longer than the astronomical day length at site 2 (figure 2*h* and table 1). When the threshold was set at 0.1 lux, the influence of the artificial light source could be detected even at site 1; day length shortened in parallel to the astronomical day length, but on some days, it became longer than 16 h (figure 2*i*). The effect of the artificial light source was also clear at site 2 and the day length was 11.83 h longer than the astronomical day length (figure 2*i* and table 1). When the threshold was set at 0.01 lux, the effect of artificial light was obvious at both sites (figure 2*j* and table 1).

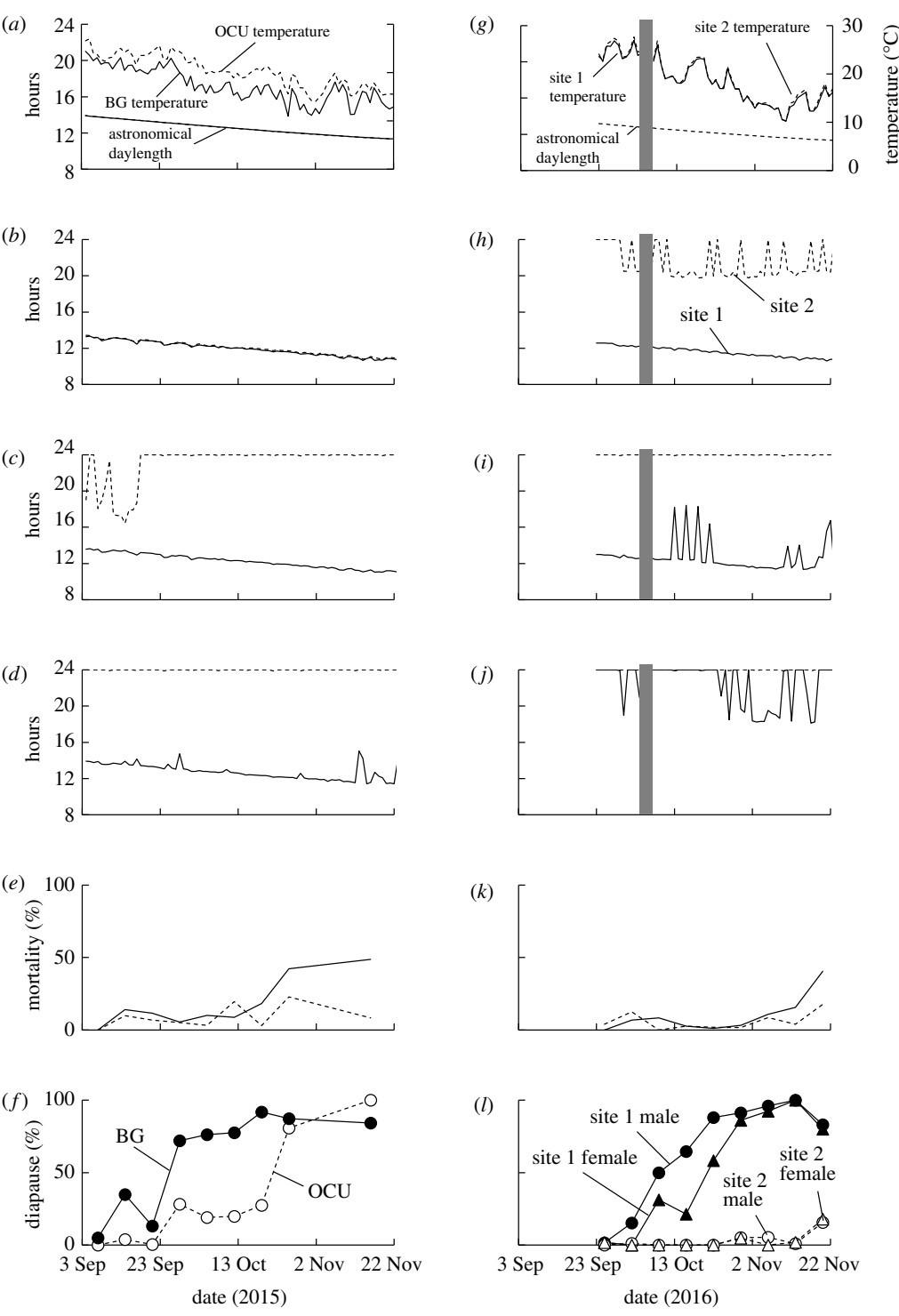

**Figure 2.** Environmental parameters, mortality and diapause incidence of *S. similis* under semi-natural conditions. (*a–f*) Experiment 1 in the Botanical Gardens (BG) located in a rural area (solid lines) and Osaka City University (OCU) located in an urban area (broken lines) from 3 September to 22 November 2015. (*g–l*) Experiment 2 at two sites (sites 1 and 2) in Osaka City University from 23 September to 22 November 2016. Site 1 (solid lines) was less affected by artificial light sources, whereas site 2 (broken lines) was severely exposed to artificial light sources. (*a,g*) Astronomical day length (dotted lines) and air temperature. (*b,h*) Period of the day of which illuminance was continuously higher than 1 lux. (*c,i*) Period of the day of which illuminance was continuously higher than 0.1 lux. (*d,j*) Period of the day of which illuminance was continuously higher than 0.01 lux. (*e,k*) Mortality. (*f*) Diapause incidences at BG (closed circles) and OCU (open circles). (*l*) Diapause incidences at site 1 (closed symbols) and site 2 (open symbols). Circles and triangles indicate males and females, respectively. Shaded areas in *g–j* indicate no data due to the incidence of a typhoon (5 and 6 October 2015).

**Table 1.** Difference (h) between the length of the astronomical day and the length of the day in each site in Experiments 1 and 2 when illuminance was continuously higher than 1, 0.1 and 0.01 lux. Data are obtained from figure 2.

| experiment | threshold illuminance (lux) | site | difference from the astronomical day length (h) |
|---|---|---|---|
| Experiment 1 | 1 | BG | $-0.56 \pm 0.12$ |
| | | OCU | $-0.48 \pm 0.11$ |
| | 0.1 | BG | $-0.25 \pm 0.10$ |
| | | OCU | $+10.7 \pm 2.57$ |
| | 0.01 | BG | $+0.20 \pm 0.55$ |
| | | OCU | $+11.5 \pm 0.77$ |
| Experiment 2 | 1 | 1 | $-0.57 \pm 0.09$ |
| | | 2 | $+9.20 \pm 1.65$ |
| | 0.1 | 1 | $+0.56 \pm 1.70$ |
| | | 2 | $+11.83 \pm 0.55$ |
| | 0.01 | 1 | $+10.35 \pm 2.18$ |
| | | 2 | $+11.83 \pm 0.55$ |

## 3.3. Temporal changes of diapause incidence under semi-natural conditions

Temporal changes in mortality and diapause incidence were assessed in Experiments 1 (figure 2*e,f*) and 2 (figure 2*k,l*). In both experiments, mortality was low until mid-October and increased from the end of October, although significant differences were undetectable (two-way ANOVA, $F = 1.18$, $p = 0.29$ in Experiment 1; $F = 0.11$, $p = 0.74$ in Experiment 2). No interaction was detected in both experiments ($F = 1.35$, $p = 0.26$ in Experiment 1; $F = 0.44$, $p = 0.52$ in Experiment 2). In Experiment 1, diapause incidences were very low at both BG and OCU on 7 September in early autumn. Diapause incidences at BG and OCU gradually increased along with the transition of days with some fluctuations; half of the flies at BG entered diapause from 21 to 28 September, whereas flies at OCU entered diapause from 19 to 26 October. Thus, there was a four-week difference between the timing of diapause of the flies between these sites (two-way ANOVA, $F = 9.71$, $p < 0.01$) with no interaction ($F = 1.97$, $p = 0.18$).

In Experiment 2, we compared the timing of diapause entry between two urban sites. We also sexed individuals. Diapause incidences were low on 25 September at both sites. Males at site 1 increased their diapause incidence gradually along with the transition of days, with half entering diapause on 9 October. Females at site 1 also increased their diapause incidence gradually, and half of the females entered diapause from 16 to 23 October. Thus, males entered diapause 7–14 days earlier than females, although no statistical difference was detected (two-way ANOVA, $F = 2.862$, $p = 0.109$). The day on which half of the population (including males and females) entered diapause ranged between 16 and 23 October. In contrast with site 1, only a few individuals had entered diapause even by 20 November in late autumn at site 2 (two-way ANOVA, $F = 42.9$, $p < 0.001$).

## 4. Discussion

*Sarcophaga similis* shows a clear photoperiodic response in diapause induction. They enter pupal diapause under short-day conditions, whereas they averted diapause under long-day conditions [35,36], as found in the present study. The response was disrupted by artificial light at night; even 0.01 lux light illumination affected diapause entry at 20°C. We also found that more females averted diapause than males under LAN conditions. It is well known that the sex ratio of overwintering individuals is heavily biased towards males not only in flesh flies [50,52,53] but also in other insects [54–58]. The present study also found that males were more inclined to enter diapause than females in the field. A previous study revealed that females are more likely to avoid diapause entry because the risks associated with overwintering are greater in females than in males [59]. By contrast, males are more likely to enter diapause, as late-emerging males are penalized in terms of fewer mating opportunities [60]. Female features that tend to avert diapause [50,61] imply that urban environments affect each sex differently. This hypothesis should be tested in a future study.

Our laboratory experiment revealed that a lower temperature enhanced diapause incidence even under the same photoperiodic conditions (both short-day and LAN conditions). In most insects with an overwintering diapause, low temperature acts to enhance diapause induction, while higher temperature acts to suppress it [62]. Owing to such a temperature effect, the negative effect of light at night was partially cancelled out at two experimental temperatures in the present study. This may have contributed to the diapause entry we observed in the field even when the habitat was polluted with artificial light at night (see below).

Our field experiment (Experiment 1) revealed that flies in the urban site delayed diapause relative to flies in the rural setting. Although we compared only two sites (BG as a rural site and OCU as an urban site) in a single year, such seasonal retardation in urban populations, possibly due to urban warming, has also been reported in other organisms [17,19,63,64]. The fly population at the urban OCU site was exposed to light at 0.1 lux or higher during the night but was still able to enter diapause in the autumn in 2 years of the experimental period. Westby & Medley [43] also found that in a mosquito, populations in urban sites are still able to enter diapause in the autumn [43]. This may be related to an ability to minimize the effect of light at night at lower temperatures, as observed in our laboratory experiment. At the urban site in Experiment 1, half of the population entered diapause between 19 and 26 October 2015, with 7-day moving temperature averages ranging from 20.5 to 18.3°C. All individuals entered diapause by 16 November and the moving average on that day was 16.8°C. A photoperiod-dependent, diapause-inducing effect of low temperature has also been reported in another flesh fly [65]. The flies did not enter pupal diapause under 13 h light/11 h dark conditions at 25°C, whereas approximately 80 and 100% of flies entered diapause under the same photoperiod at 20 and 15°C, respectively. By contrast, most flies averted diapause under 15 h light/9 h dark conditions, irrespective of temperature [65]. Although the effects of temperature on diapause induction at light-polluted sites have not been investigated in any other insect species, such a photoperiod-dependent low-temperature effect would contribute to diapause induction at light-polluted urban sites.

Experiment 2 also revealed that flies facing an urban residential area failed diapause entry, although flies at the less-polluted urban site successfully entered diapause. It is important to note that the illuminance level at our experimental site (site 2 in Experiment 2) during nights was approximately 7 lux, which is an ordinal level of lighting in a residential area (Japan Ministry of the Environment, www.env.go.jp). Illuminance, not temperature, is the main difference between these sites. These results suggest that high levels of artificial light at night disrupt the seasonal adaptation of the flesh fly. However, it is necessary to undertake similar experiments at multiple urban sites for multiple years to draw a general conclusion because we compared only two sites in a single year in the present study. Studies focusing on the effects of artificial light at night on seasonal biological events are limited [13]. Additionally, such a strong effect on seasonality has been assumed conceptually [13,66] but has been poorly tested empirically. Furthermore, most of the studies in this context used 'simulated' artificial light at night to investigate its effect on seasonality [26,30,31,43,44,67–69]. Field studies with 'natural' artificial light at night are awaited [70,71]. Although we obtained an extreme dataset in site 2 in Experiment 2, possibly due to persistent night-time-long fluorescent light emissions, other forms of artificial lighting found in other urban sites (from short pulses to persistent long emissions, from narrow to broad spectra, from low to high emission intensities, and from locally focused illumination to more pervasive illumination) may have more or less serious consequences [13].

In the present study, we provide an example with the negative effect of residential artificial light at night on diapause entry and the positive effect of low temperature on diapause induction in the flesh fly. It is important to note that our findings may not be directly applicable to other insect species inhabiting other urban/rural sites. *Sarcophaga similis* is highly sensitive to light at night (as low as 0.01 lux). This is one of the lowest thresholds among insects [41], and thus, insect species with low sensitivity may be less affected by artificial light at night. Spectral sensitivity in the photoperiodic response also varied among insects [72], and different forms of artificial lighting have unique spectral signatures [13]. The effects of temperature on the photoperiodic response also varied among insect species [62]. Especially, constant temperature, temperature cycles, pulses and steps play distinct roles in the photoperiodic response. Diurnal and seasonal patterns of urban warming are quite variable among cities [73]. In the present study, we demonstrated a series of field experiments in Osaka, the third-largest city in Japan, which is located in the warm-temperate region. Urban warming and artificial light at night are not only spatially and temporally but also qualitatively and quantitatively distinct among urban/rural sites in different cities [3,13,42,73]. Future studies with multiple insect species at multiple sites in multiple cities in different climatic regions would clarify what levels of light pollution and urban warming affect insect seasonal adaptation.

Data accessibility. Data are available from the Dryad Digital Repository: https://doi.org/10.5061/dryad.nk98sf7t3 [74].
The data are provided in electronic supplementary material: https://doi.org/10.6084/m9.figshare.14921589 [75].

Authors' contributions. S.G.G. conceived the study, designed the study, coordinated the study, helped draft the manuscript and critically revised the manuscript; A.M. participated in the design of the study, collected laboratory and field data, carried out the statistical analyses, drafted the manuscript and critically revised the manuscript; K.Y. collected field data. All authors gave final approval for publication and agree to be held accountable for the work performed therein.

Competing interests. The authors declare no competing interests.

Funding. This study was supported in part by an Osaka City University Strategic Research Grant 2015–2017 for basic research on SGG and by JSPS KAKENHI grant no. 16K08101 & 19H02971.

Acknowledgements. This study is based on the preliminary research performed by Jun Tagaya and Fumi Kitamura at Osaka City University. We thank Prof. Dr Moritoshi Iino, the Principal of the Botanical Gardens, Graduate School of Science, Osaka City University, Japan, and garden members for allowing the use of the gardens for this study. We also thank Dr Sakiko Shiga, Osaka University, for providing us with her Toyonaka flesh fly colony. We acknowledge Dr Koichi Soga, Osaka City University, for allowing us to use the colour rendering illuminometer. We thank Dr Takako Shizuka at Osaka City University for preparing figures. We also acknowledge Editage (www.editage.com) for English corrections.

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
