## [Peer Review File · Royal Society Open Science]

Review History

Decision letter (RSOS-210866.R0)

Dear Dr Mukai

On behalf of the Editors, we are pleased to inform you that your Manuscript RSOS-210866 "Urban warming and artificial light alter dormancy in the flesh fly" has been accepted for publication in Royal Society Open Science subject to minor revision in accordance with the referees' reports. Please find the referees' comments along with any feedback from the Editors below my signature.

Please submit your revised manuscript and required files (see below) no later than 7 days from today's (ie 04-Jun-2021) date. Note: the ScholarOne system will 'lock' if submission of the revision

is attempted 7 or more days after the deadline. If you do not think you will be able to meet this deadline please contact the editorial office immediately.

on behalf of Professor Andrew Simons (Associate Editor) and Pete Smith (Subject Editor)
openscience@royalsociety.org

Associate Editor Comments to Author (Professor Andrew Simons):

Comments to the Author:

The authors have addressed most of the concerns of the original, mostly positive, reviews.

However, a fuller response to the major concern of both referees is still needed. Specifically, because the first field experiment used only one rural and one urban environment, and the second is based again on just two (urban) environments, it is impossible to draw generalizations about urban and rural environments, and inferences are limited to the specific sites themselves.

This imposes a major limitation on inferential power of this study, and the authors should insert a cautionary statement of this important caveat in the Discussion section alongside the discussion of results of both field experiments. The edits to the final paragraph points out that there is a problem, but is not sufficient because it does not specify what the problem is, or how it limits inference.

===PREPARING YOUR MANUSCRIPT===

While not essential, it will speed up the preparation of your manuscript proof if you format your references/bibliography in Vancouver style (please see

<https://royalsociety.org/journals/authors/author-guidelines/#formatting>). You should include DOIs for as many of the references as possible.

===PREPARING YOUR REVISION IN SCHOLARONE===

<https://royalsociety.org/journals/authors/author-guidelines/#data>. You should ensure that you cite the dataset in your reference list. If you have deposited data etc in the Dryad repository,

please only include the 'For publication' link at this stage. You should remove the 'For review' link.

Author's Response to Decision Letter for (RSOS-210866.R0)

See Appendix A.

Decision letter (RSOS-210866.R1)

Dear Dr Mukai,

I am pleased to inform you that your manuscript entitled "Urban warming and artificial light alter dormancy in the flesh fly" is now accepted for publication in Royal Society Open Science.

You can expect to receive a proof of your article in the near future. Please contact the editorial office (openscience@royalsociety.org) and the production office (openscience_proofs@royalsociety.org) to let us know if you are likely to be away from e-mail contact -- if you are going to be away, please nominate a co-author (if available) to manage the proofing process, and ensure they are copied into your email to the journal. Due to rapid

publication and an extremely tight schedule, if comments are not received, your paper may experience a delay in publication.

on behalf of Professor Andrew Simons (Associate Editor) and Pete Smith (Subject Editor)
openscience@royalsociety.org

Appendix A

Responses to the editor

Associate Editor Comments to Author (Professor Andrew Simons):

Comments to the Author:

The authors have addressed most of the concerns of the original, mostly positive, reviews. However, a fuller response to the major concern of both referees is still needed. Specifically, because the first field experiment used only one rural and one urban environment, and the second is based again on just two (urban) environments, it is impossible to draw generalizations about urban and rural environments, and inferences are limited to the specific sites themselves. This imposes a major limitation on inferential power of this study, and the authors should insert a cautionary statement of this important caveat in the Discussion section alongside the discussion of results of both field experiments. The edits to the final paragraph points out that there is a problem, but is not sufficient because it does not specify what the problem is, or how it limits inference.

(Response)

According to this editor's comment, we described our experimental design and the limitation of our experiments in the Discussion section.

For Experiment 1 (comparison between one rural site and one urban site), we stressed our experimental design as follows (the third paragraph in the Discussion section): Although we compared only two sites (BG as a rural site and OCU as an urban site) in a single year, such seasonal retardation in urban populations, possibly due to urban warming, has also been reported in other organisms.

For Experiment 2 (comparison between 2 urban sites), we described the limitation of this study as follows (the fourth paragraph in the Discussion section): However, it is necessary to undertake similar experiments at multiple urban sites for multiple years to draw a general conclusion because we compared only 2 sites in a single year in the present study.

In addition, we replaced the last paragraph describing the ecological consequence of low diapause incidence in autumn with a new paragraph that describes variations of insect photoperiodism and urban/rural environmental conditions, to show what the problem of this study is and how it limits inference.